# Artificial Intelligence for Predicting the Aesthetic Component of the Index of Orthodontic Treatment Need

**DOI:** 10.3390/bioengineering11090861

**Published:** 2024-08-23

**Authors:** Leah Stetzel, Florence Foucher, Seung Jin Jang, Tai-Hsien Wu, Henry Fields, Fernanda Schumacher, Stephen Richmond, Ching-Chang Ko

**Affiliations:** 1Division of Orthodontics, The Ohio State University, 305 W. 12th Avenue, Columbus, OH 43210, USA; 2Division of Biostatistics, The Ohio State University, 1841 Neil Avenue, Columbus, OH 43210, USA; 3Department of Orthodontics, Cardiff University, Heath Park, Cardiff CF14 4XY, UK

**Keywords:** Index of Orthodontic Treatment Need, aesthetic component, artificial intelligence

## Abstract

The aesthetic component (AC) of the Index of Orthodontic Treatment Need (IOTN) is internationally recognized as a reliable and valid method for assessing aesthetic treatment need. The objective of this study is to use artificial intelligence (AI) to automate the AC assessment. A total of 1009 pre-treatment frontal intraoral photos with overjet values were collected. Each photo was graded by an experienced calibration clinician. The AI was trained using the intraoral images, overjet, and two other approaches. For Scheme 1, the training data were AC 1–10. For Scheme 2, the training data were either the two groups AC 1–5 and AC 6–10 or the three groups AC 1–4, AC 5–7, and AC 8–10. Sensitivity, specificity, positive predictive value, negative predictive value, and accuracy were measured for all approaches. The performance was tested without overjet values as input. The intra-rater reliability for the grader, using kappa, was 0.84 (95% CI 0.76–0.93). Scheme 1 had 77% sensitivity, 88% specificity, 82% accuracy, 89% PPV, and 75% NPV in predicting the binary groups. All other schemes offered poor tradeoffs. Findings after omitting overjet and dataset supplementation results were mixed, depending upon perspective. We have developed deep learning-based algorithms that can predict treatment need based on IOTN-AC reference standards; this provides an adjunct to clinical assessment of dental aesthetics.

## 1. Introduction

The National Health Service has been facing severe pressure to reduce costs due to consequences of the COVID-19 pandemic, chronic understaffing issues, and a fiscal deficit [1]. Yet, a recent survey in May of 2021 of members of the British Orthodontic Society showed a marked increase in demand for orthodontic services [2]. NHS spending on primary care orthodontic services is approximately GBP £250 million annually. (£: Great British Pound) [3]. It is increasingly important to distribute limited funds in a manner such that those in need of treatment are eligible for and obtain orthodontic services.

The importance of a smile is widely accepted, not only by society but also by the scientific literature. In a study by Shaw [4], it was found that children’s dental features affect a viewer’s perceptions of their attractiveness and personal characteristics such as intelligence and aggressiveness. Similar results were confirmed by Papio et al. [5] in the adult population. Another study, in 2011 [6], found that ratings of attractiveness, intelligence, conscientiousness, agreeableness, and extraversion differed significantly depending on dental relationships or occlusion. Subjects with normal occlusion were rated the most positively in these categories. Because of this evidence, orthodontic treatment to improve esthetics and related social, intellectual, and integrity-based judgements is sought by patients and also recommended by orthodontists.

One of the most widely used assessments of orthodontic treatment need is the Index of Orthodontic Treatment Need (IOTN). Multiple studies have verified the reliability of IOTN and supported its use with international populations [7,8,9]. IOTN has two components: the dental health component (DHC) and the aesthetic component (AC).

The DHC consists of a five-point scale based on occlusal traits such as missing teeth, crossbites, displacement of contact points, overjet, and overbite, where Grade 1 signifies “no treatment need” and Grade 5 signifies “great treatment need”. The AC consists of a 10-point scale illustrated by a series of photographs that represent different levels of dental attractiveness [10]. In utilizing the IOTN-AC, a rating of 1–10 is assigned for *overall* dental attractiveness rather than particular similarities to the photographs. The final value should reflect treatment need on the grounds of esthetic impairment and, consequently, the psychosocial need for orthodontic treatment [11].

With the use of a validation exercise, Richmond et al. [8] reported that IOTN-AC grades could be partitioned into three treatment-need subgroups: no need, borderline need, and definite treatment need. These reflect AC grades 1–4, 5–7, and 8–10, respectively, in this modified grouping [12].

IOTN is currently used by the National Health Service (NHS) to determine whether children qualify for covered orthodontic treatment. Patients with an IOTN-DHC of 4 or 5 are eligible for NHS orthodontic treatment. However, the decision on treatment for borderline malocclusions, such as those with DHC of 3, is known to be difficult [13,14]. In 2006, a prioritization system was introduced so that these borderline cases (DHC = 3) required an AC grade of 6 or more in order to receive eligibility for treatment within the NHS [15]. It is clear that the AC evaluation impacts the ability of patients to receive care. 

With increasing demand for orthodontic care, reducing the workload of orthodontists and increasing at-home patient assessments are appealing ideas. The use of artificial intelligence (AI) has aided both the medical and the dental fields in diagnosis automation [16,17,18,19]. Orthodontics may be one of the dental specialties earliest in adapting AI into its practice [20] A systematic review and meta-analysis conducted in 2021 seeking to examine the accuracy of deep learning (a branch of AI which utilizes neural networks) in detecting landmarks on cephalometric radiographs demonstrated relatively high accuracy [21]. Others noted that deep learning performed similar to seasoned clinicians, and perhaps even better than inexperienced ones [22,23].

AI’s ability to assess orthodontic treatment needs has been explored. In a study by Murata et al. [24], AI was able to classify patients into five orthodontic treatment-need categories with 45% accuracy. The categories ranged from 1 (no need for treatment) to 5 (need for treatment) and were based on intraoral images taken from five different angles.

In the current study, we propose the use of artificial intelligence (AI) to augment the IOTN-AC assessment, which would allow for more objective diagnoses, a reduced workload for orthodontists, at-home patient assessments, and potential utilization by third-party payers. The purpose of this study was to collect a dataset of patients’ frontal intraoral images with the corresponding IOTN-AC classification and overjet and develop a deep-learning based AI algorithm that could identify the IOTN-AC.

## 2. Materials and Methods

### 2.1. Data Collection

A total of 1009 intraoral images were gathered in a quota-sampling manner, such that they mirrored the U.S. population according to race and overjet values, the latter established from the epidemiological literature [25,26,27]. The 1009 intraoral images with a corresponding median overjet were assessed by an experienced calibration examiner. Each photo was assigned an AC score. After a two-week wash-out period, 200 randomly chosen images were tested for reliability using kappa. The 1009 images then served as our gold standard.

### 2.2. Deep Learning

We developed a deep neural network, called the IOTN network, which takes two inputs and has three modules. The inputs are a 2D frontal intraoral image and an overjet numeric value, which was the median value of the overjet range at central incisors. The modules consist of a convolutional neural network (CNN) module, an overjet module, and an output module, corresponding to the two inputs (Figure 1).

In the CNN module, we used Residual Network 34 (ResNet34), the most widely used neural network in both computer vision and medical imaging, to be the backbone used to extract 20 hidden features [28]. In the overjet module, a two-layer fully connected network with the hyperbolic tangent activation function was used to learn 4 hidden features in an abstract domain from the overjet value. The 20 CNN features and 4 hidden overjet features were concatenated and fed into the final classification module to output the prediction. The output module is comprised of two fully connected layers followed by the hyperbolic tangent activation function. 

It is worth noting that we consider this supervised task a regression problem instead of a classification problem, since the IOTN grade implies the severity of the patient’s oral conditions. For regression problems, the most used activation functions are the hyperbolic tangent function and the sigmoid function. Whereas for classification problems, the SoftMax function is the most used. The IOTN classification system uses integer numbers to represent treatment need, implying an ordinal relation. For example, patients with IOTN 1 (little to no need for treatment) look more similar to those patients with IOTN 5–7 (borderline need for treatment) than to patients with IOTN 8–10 (great need for treatment). Therefore, the hyperbolic tangent activation function was adopted as the last layer in the output module, instead of the SoftMax activation function. Furthermore, because the IOTN-AC grades are equidistant from each other, classifications can be considered interval data.

### 2.3. Implementation

After collection of the 1009 photos, the IOTN network was trained, validated, and tested in a supervised learning manner. In machine learning, multiple models are often considered (or “trained”) before a final model is chosen (or “validated”). The validated model is the one most optimized in terms of network parameters. The chosen, or validated, model is then “tested” with new, never-before-seen data in order to evaluate its performance and generalizability to unseen data [29].

Of the 1009 gathered, 800 images were used in the training phase, 40 images were used in the validation phase, and 200 were used in the testing phase. In the training phase, three inputs were given to the network: (1) an intraoral image, (2) an overjet value, and (3) the gold standard (via the loss function, a measure of the difference between the gold standard and prediction). The discrepancy between the gold standard and the prediction was back-propagated to each layer of the network to update their parameters. Figure 2 shows a schematic displaying how the IOTN network was trained.

To test the model, the AI was given 200 unique new images with corresponding OJ values and was tasked with grading the IOTN-AC. Figure 3 shows a schematic of the testing phase. The testing dataset mirrored the IOTN distribution of our representative sample of 1009. That is, the testing data had the same percentage of each IOTN-AC grade as indicated in the initial data collection. Sensitivity (SEN) is the proportion of correct AI gradings for IOTN-AC images that were deemed as needing treatment by the gold standard. Specificity (SPE) is the proportion of correct AI gradings for images that were deemed as not needing treatment. Positive Predictive Value (PPV) is the proportion of AI gradings indicating a need for treatment that were correct. Negative Predictive Value (NPV) is the proportion of AI gradings indicating no need for treatment that were correct. Accuracy (ACC) is the overall proportion of correct AI gradings compared to the gold standard.

### 2.4. Data Augmentation and Transfer Learning

To avoid overfitting (when training results exceed those for novel data) in our relatively small dataset, we adopted two techniques: data augmentation and transfer learning. For data augmentation, we randomly applied different image filters on each image to “create” different images from the same source. The image filters used in this study include cropping and padding, sharpening, embossing, Gaussian noise, Gaussian blur, contrast adjustment, and dropout (i.e., randomly removing some pixels). Each filter had a random chance of being applied on the training images. By performing this heavy augmentation configuration, we expanded our training data to 200 images for each grade, for a total of 2000 images. It is important to note that although each grade was augmented in order to have 200 images, the image diversity of these grades was not equal.

The second technique we applied was transfer learning, which is the process of applying previously acquired knowledge to new situations. This technique has been widely used in medical imaging studies since it is difficult to collect a large number of novel medical images. The pre-trained parameters of the ResNet34, previously trained by ImageNet (an open dataset containing 1,281,167 training natural images, 50,000 validation natural images, and 100,000 test natural images) for 1000-object classification, were used in our CNN module. Due to this, our CNN module had an excellent initial ability to extract and recognize abstract features from intraoral photos since the network already could recognize those natural images in ImageNet dataset. Then, we applied our augmented intraoral images to fine-tune the CNN module as to its ability to predict IOTN. All the implementations were accomplished with Pytorch, an open-source deep learning library with the Python programming language [30]. The data augmentation was carried out by imgaug, a library for image augmentation in machine learning experiments. The pre-trained ResNet34 was downloaded from Pytorch.

Scheme 0

In the training phase of our first scheme, denoted as Scheme 0, the gold standard was IOTN 1–10. In the testing phase of Scheme 0, the IOTN network predicted an IOTN-AC grade 1–10 for each image. 

2.Scheme 1

In the training phase of Scheme 1, the same training configuration as Scheme 0 was used, in which the gold standard was IOTN-AC 1–10. In the testing phase, however, we added a procedure, called mapping, at the end to simplify the IOTN-AC prediction and gold standard into binary or ternary classes. In the binary classification, IOTN-AC 1–5 was simplified to I, and IOTN-AC 6–10 was simplified to II. In the ternary classification, IOTN-AC 1–4 was simplified to I, IOTN-AC 5–7 was simplified to II, and IOTN-AC 8–10 was simplified to III. 

3.Scheme 2

In the training phase of Scheme 2, the gold standard was simplified into the binary and ternary groupings, as described above. In Scheme 2, the IOTN network automatically predicted the simplified binary and ternary classifications, and mapping was unnecessary.

A summary of the schemes’ trainings and tests can be found in Figure 4.


*The IOTN Network Variant and Supplemented Dataset*


In addition, we also developed an IOTN network variant which only takes the intraoral image as input (i.e., removes the overjet module in the original IOTN network). 

To further test the influence of the size of the dataset on the overall AI system performance, 64 more intraoral images previously graded using the IOTN-AC were obtained from Dr. Richmond. The IOTN network was trained and tested again using Scheme 1.

### 2.5. Statistical Analysis

All schemes’ performances were measured by calculating sensitivity (SEN), specificity (SPE), positive predictive value (PPV), negative predictive value (NPV), accuracy (ACC), and balanced accuracy (BA) [31,32].

For the binary predictions, an IOTN-AC of 6–10 was considered a “positive” test and prediction, while an IOTN-AC of 1–5 was considered a “negative” test and prediction. 

For the ternary prediction, sen, spec, PPV, and NPV were calculated for each treatment-need group, I-III. For example, for the treatment-need group III, a true positive was when the actual treatment-need group was III and the predicted treatment-need group was III. A false positive was when the actual treatment-need group was either I or II and the predicted treatment-need group was III. A true negative was when the actual treatment-need group was either I or II and the predicted treatment-need group was either I or II. Finally, a false negative was when the actual treatment-need group was III and the predicted treatment-need group was I or II. 

Similarly, for the prediction of IOTN 1–10, sen, spec, PPV, and NPV were calculated for each individual grade.

## 3. Results

The gold-standard IOTN grader demonstrated excellent intra-rater reliability in the identification of IOTN grades 1–10, as tested using kappa agreement, where the weighted kappa was 0.84 (95% CI, 0.76 to 0.93).

The results of our initial data collection provided a representation of IOTN-AC grades in the U.S. population. The most infrequent IOTN-AC grade was IOTN 1, which represented 1% of our sample. IOTN-AC 9 and IOTN 10 were also uncommon, and each represented 3% of our sample. IOTN-AC 6 and 7 were the most frequent grades in our sample, representing 20% and 18%, respectively. A complete IOTN-AC distribution for our sample can be found in Table 1.

### 3.1. Prediction of IOTN-AC 1–10

For predicting IOTN-AC 1–10, Scheme 0 had poor values for sensitivity, positive predictive value, and accuracy. When analyzing the performance of Scheme 0, 89% of errors (or values for absolute difference between gold standard and prediction, when >0) were of either 1 or 2. 

### 3.2. Prediction of IOTN-AC 1–5 (I) and 6–10 (II)—Binary

For the binary predictions, Scheme 1 outperformed Scheme 2 in sensitivity, specificity, positive predictive value, negative predictive value, and accuracy. Scheme 1 was able to identify images with IOTN-AC 6–10 77% of the time. The results of the binary predictions for Scheme 1 and Scheme 2 can be seen in Figure 5.

### 3.3. Prediction of IOTN-AC 1–4 (I), 5–7 (II), and 8–10 (II)—Ternary

For the ternary predictions, on average, Scheme 1 outperformed Scheme 2 in all aspects: sensitivity, specificity, positive predictive value, negative predictive value, and accuracy. 

When analyzing the outcomes for each prediction group, it is evident that Scheme 1 misclassified actual “Borderline Need” subjects into both the “No Need” and “Great Need” categories, whereas Scheme 2 mis-predicted actual “Borderline Need” subjects by placing them into only the “No Need” category. Scheme 2 had substantially low sensitivity, and substantially high specificity and PPV. In this case, Scheme 2 mis-predicted all but one of the actual “Great Need” subjects, placing them into the “Borderline Need” group instead. Furthermore, there were no false positives for “Great Need” in Scheme 2. In both Scheme 1 and Scheme 2, the “Borderline Need” group had the highest sensitivity and the lowest specificity, compared to both “No Need” and “Great Need” groups. These results are visualized in Figure 6.

### 3.4. Predictions without Overjet and with Supplemented Data

Without overjet, the model’s performance decreased in every metric, on average, for the ternary predictions. For the binary predictions, specificity and positive predicative value increased, while every other metric decreased.

### 3.5. Predictions with Sample Size and Augmented Data

The model’s accuracy was positively correlated with the increase in sample size; however, accuracy was substantially improved when up-sampling and image augmentation were implemented, increasing from 56% to 65% at a 25% sampling size. When increasing the sample size with image augmentation alone, the accuracy was saturated at 75%, but decreased by 15% at the end; this could be explained by the inability of the software to recognize the images once they were excessively tweaked. The results of the binary and ternary predictions with our supplemented data can be found in Figure 7. 

### 3.6. Summary

The performance measures for all the schemes can be found in Table 2.

## 4. Discussion

In this study, we proposed the use of artificial intelligence to augment the IOTN-AC assessment. We proposed multiple schemes of training and testing, and it is clear that the results are variable, depending upon how the AI model is trained and tested. 

The experienced calibration examiner had nearly perfect intra-rater reliability by weighted kappa, according to the test described in Cohen [33]. This served as a strong underpinning for the study.

When originally attempting to classify the specific need-categories of 1–10, our model (Scheme 0) proved inaccurate (acc = 34%). However, when analyzing the discrepancies, or error, in this model, it was noted that 89% of errors were of only 1 or 2 grades, and a positive correlation was found (r = 0.74). It is well known that classification problems become more challenging as the number of classes increases, and a recent study suggests that this increased complexity is due, at least in part, to the heterogeneity in decision boundaries [34].

In order to improve our results, Scheme 1 and Scheme 2 were developed, and artificial intelligence was tasked to identify the broader treatment-need categories (binary and ternary classifications). Emphasis was given to predicting these broader treatment-need categories, due to their practicality. The binary classification system is especially promising among those 18 years or younger and enrolled in the NHS. If one is considered borderline in the DHC, the binary IOTN-AC classification can determine whether you are eligible for NHS-funded treatment (IOTN 6–10) or if you will be ineligible (IOTN 1–5). The ternary classification is more descriptive, in which IOTN 1–4 indicates little to no treatment need, IOTN 5–7 indicates moderate treatment need, and IOTN 8–10 indicates great treatment need, but is less useful in real application.

Certain metrics lead to the conclusion that Scheme 1 outperforms Scheme 2, and that Scheme 1 shows promise. The value of the outcomes really is one of perspective. If you are the payer, you do not want false positives, so high specificity and high PPV are critical. In fact, given the need to conserve funds for either the government or the administrator as net profit, you do not care about false negatives.

From a patient’s or a provider’s viewpoint, it is undesirable to receive false negative scores. So, high sensitivity and high NPV are most important. You want all who deserve it to be funded, and do not care if there are some false positives, because all who qualify (and then some, possibly) will be funded. Our judgement is that with public funds, shortchanging those who qualify is worse than mistakenly funding a few extra cases in error. All deserving cases will be provided with funding.

It is important to note that certain third-party payers, such as the NHS, are funded by the public. According to the NHS Constitution for England, Principles #2 and #6, “Access to NHS services is based on clinical need”, and the NHS “is committed to providing the most effective, fair and sustainable use of finite resources” [35]. Therefore, Scheme 1 with binary prediction could be considered promising for use by the NHS, after further improvement.

The ternary predictions may be clinically useful as they are more descriptive than the binary predictions. However, due to the poor sensitivity of the “Great Need” category, if this model were to be used to determine eligibility for care, many patients with true “Great Need” for treatment would be mis-categorized as “Borderline Need”. This may lead to an excess of appeals to third-party payers.

When analyzing the binary grouping results (which is necessary when a patient has a DHC = 3 in the NHS) of Scheme 1 vs. Scheme 2, Scheme 1 performs better overall. It would be desirable to have an automated system that can generate minimal false negatives (high sens), so that all of those needing treatment are captured.

Overall, the results of Scheme 1 were more promising than those of Scheme 2 when considering both binary and ternary predictions. Therefore, we decided to investigate how Scheme 1 would perform without overjet input. This would allow for less clinical error and less variation among practitioners. In the binary classification, ACC, SEN, and NPV increased by 1%, 3%, and 2%, respectively. There was a decrease in SPE, while PPV stayed the same. In the ternary classification, all values decreased slightly, while NPV improved. This slight decrease in results may not be clinically significant. 

We investigated how increasing the sample or training sizes could impact our results. By supplementing our dataset with an additional 64 images, we were able to ensure that each treatment-need category had at least 20 images. In our original dataset (which represented the U.S. population), IOTN grades 1, 9, and 10 were significantly under-represented. When the data were supplemented, the prevalence across classifications was increased, especially for IOTN 1, 9, and 10. Supplementing the dataset in this manner improved our results, and it can be assumed that further increasing the sample size would further improve our results. Also, with an increase in prevalence, one would expect to see an increase in positive predictive value and a decrease in negative predictive value, and this was observed in the binary predictions. Again, this was not in the interest of the patients.

Limitations of this study include the fact that while our AI system demonstrated quicker results compared to the manual method, it did not achieve acceptable accuracy. This limitation is considerable, as the typical threshold for acceptable accuracy in clinical dentistry and orthodontics is 90% or higher [36,37,38]. Our current model falls short of this benchmark, indicating the need for further refinement.

One significant challenge is that the current methodology may not fully leverage the potential of our AI system. There could be a discrepancy in how images are assessed; the AI may have failed to produce reliable results because it could not effectively capture the holistic approach that human observers use. Teeth can be viewed as a whole unit or a set of features. Understanding whether discrete changes in images correspond to a grading of severity or if severity is established more holistically by observers is crucial. Our current algorithm may not have effectively balanced these approaches, and future work should explore this aspect in greater depth.

Another limitation of our study is the relatively small sample size of approximately one thousand images, which may affect the robustness of our AI algorithm in diverse clinical scenarios. While our sample was designed to mirror the U.S. population according to race, the potential differences in algorithmic errors across different racial groups were not specifically investigated. The variability in gingival color among races might influence the performance of the AI model [39]. Investigating these differences is necessary to improve the algorithm’s accuracy and fairness across diverse populations [40]. Additionally, the variability in image quality and the need for image manipulation may introduce errors. Standardized photography protocols could help reduce these errors by minimizing the need for extensive image augmentation. Implementing consistent imaging techniques might enhance the algorithm’s accuracy and reliability. Furthermore, plans for future work include integrating this algorithm with image segmentation techniques to separate the teeth from the gingiva in intraoral photos, mitigating the influence of gingival color on the algorithm’s predictions. Future studies with larger and more diverse datasets are necessary to thoroughly evaluate these aspects and to determine the impacts of racial differences and standardized photography on algorithm accuracy.

In summary, while our study provides insights into developing an AI solution for the aesthetic component of the IOTN, further work remains in order to achieve the required accuracy for clinical usability. Future studies should focus on refining the algorithm, balancing the training dataset, and exploring both holistic and analytic image assessment approaches to enhance the system’s reliability and applicability in a clinical setting.

## 5. Conclusions

We have developed deep learning-based algorithms capable of predicting dental aesthetic needs based on IOTN-AC reference standards. Our approach, using the AC 1–10 scale input with binary testing, proved superior compared to other AC categorizations, aligning well with patient-centered public policy perspectives. While our AI system has shown promising results, achieving a level of accuracy that is not yet sufficient for clinical use highlights the potential for further refinement. Specifically, the elimination of overjet enhanced and simplified our results, demonstrating potential improvements in accuracy and clinical applicability. Further studies should focus on refining the algorithm and exploring holistic versus analytic approaches to image assessment in order to enhance reliability in clinical settings.

## Figures and Tables

**Figure 1 bioengineering-11-00861-f001:**
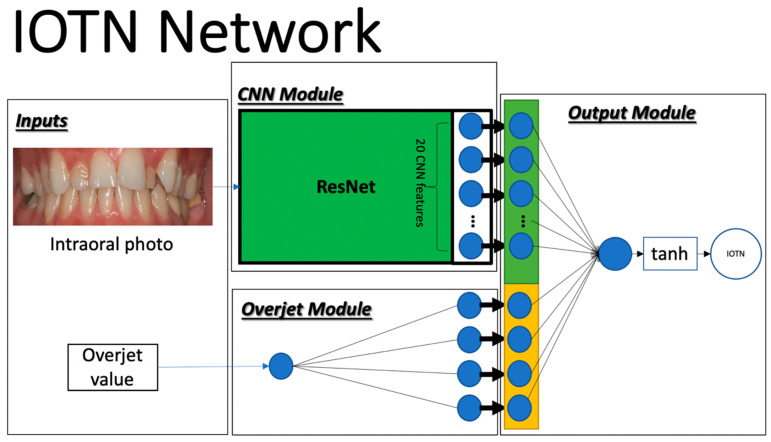
The IOTN network with 2 inputs, an overjet module, a CNN module, and an output module.

**Figure 2 bioengineering-11-00861-f002:**
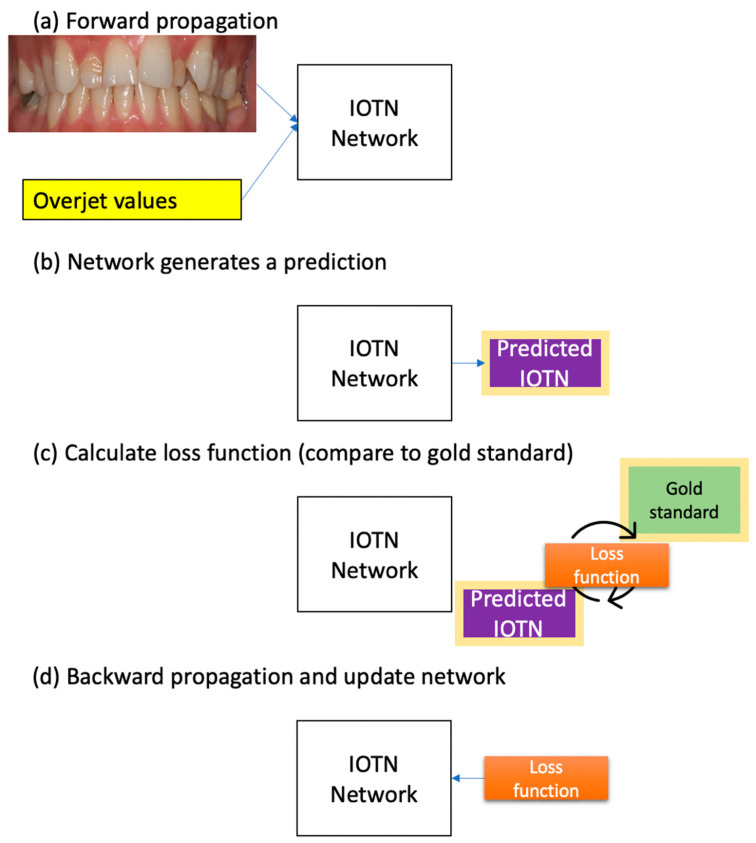
Schematic of a training process. (**a**) Forward propagation: The model receives the first two inputs, namely the intraoral image and the overjet value. (**b**) Network generates a prediction: The network generates a prediction based on these inputs, aiming to learn the output of the IOTN. (**c**) Calculate loss function: The predicted IOTN value is compared to the gold standard (third input) to calculate the discrepancy. (**d**) Backward propagation and update network: The discrepancy is back propagated through each layer of the network to update their parameters.

**Figure 3 bioengineering-11-00861-f003:**
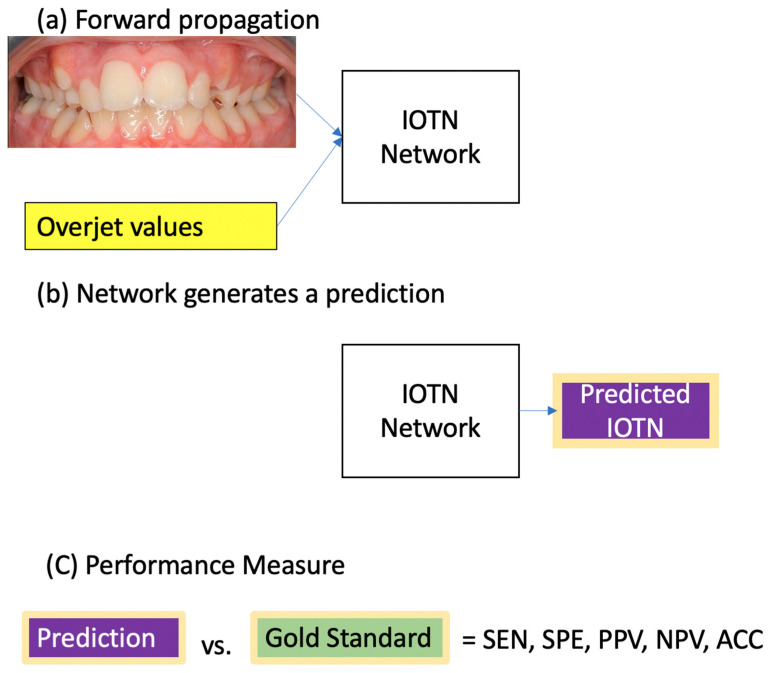
Schematic of a test process. (**a**) Forward propagation: The model takes the two inputs, namely the intraoral image and the overjet value. (**b**) Network generates a prediction: The network predicts an IOTN-AC grade based on these inputs. (**c**) Performance Measurement: Performance is assessed by comparing the predicted value to the gold standard using diagnostic metrics, including SEN (sensitivity), SPE (specificity), PPV (positive predictive value), NPV (negative predictive value), and ACC (accuracy).

**Figure 4 bioengineering-11-00861-f004:**
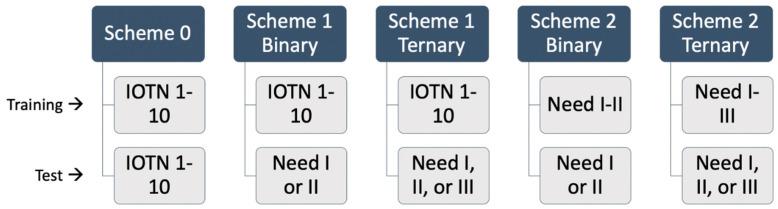
Summary of Scheme 0, Scheme 1, and Scheme 2: training and test. Need I or II is binary (IOTN 1–5 and 6–10). Need I, II, or III is ternary (IOTN 1–4, 5–7, and 8–10).

**Figure 5 bioengineering-11-00861-f005:**
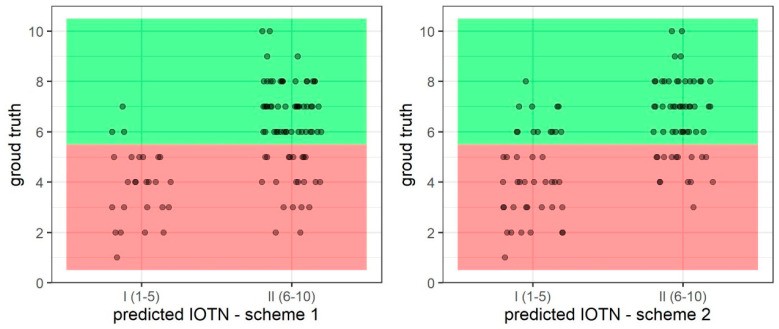
Scatter plots depicting binary prediction results for Scheme 1 (**left**) and Scheme 2 (**right**). In these plots, red region denotes IOTN-AC grades 1–5, while green region denotes IOTN-AC grades 6–10, as per the calibration clinician. Correct classifications are represented in the lower left and upper right quadrants of the plots.

**Figure 6 bioengineering-11-00861-f006:**
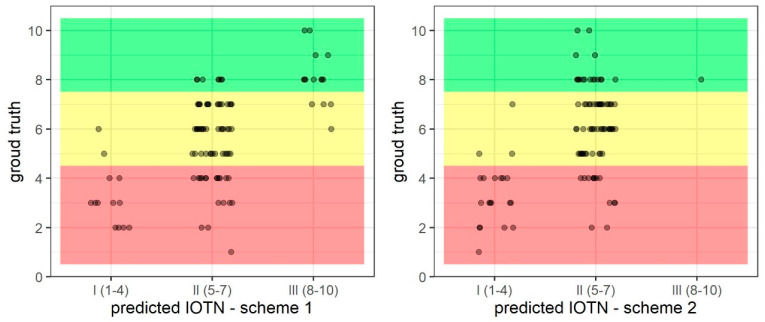
Scatter plots for ternary prediction results of Scheme 1 (**left**) and Scheme 2 (**right**). In these plots, red region denotes IOTN-AC grades 1–4, yellow region denotes IOTN-AC grades 5–7, and green region denotes IOTN-AC grades 8–10, as per the calibration clinician. Outcomes in the lower left, center, and upper right regions are correct classifications.

**Figure 7 bioengineering-11-00861-f007:**
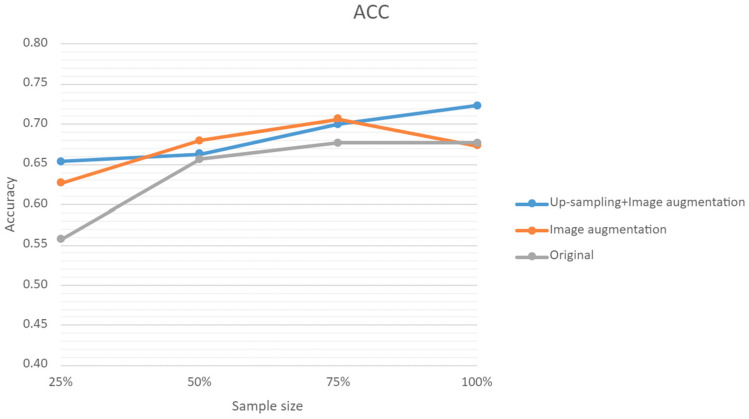
Line graph representing the accuracy of all schemes.

**Table 1 bioengineering-11-00861-t001:** IOTN-AC grades for our sample, which was selected in such a manner as to be representative of the IOTN-AC grades in the U.S. population.

IOTN	Count	Percentage
1	7	1%
2	49	5%
3	97	10%
4	134	13%
5	149	15%
6	203	20%
7	182	18%
8	125	12%
9	31	3%
10	32	3%

**Table 2 bioengineering-11-00861-t002:** Performance measures of the schemes.

Scheme	Sens	Spec	PPV	NPV	Acc
Scheme 0	0.27	0.92	0.50	0.92	0.34
Scheme 1 Binary	0.77	0.88	0.89	0.75	0.82
Scheme 2 Binary	0.76	0.87	0.88	0.74	0.81
Scheme 1 Ternary	0.65	0.83	0.77	0.85	0.72
Scheme 2 Ternary	0.63	0.81	0.67	0.82	0.67
Scheme 1 w/out OJ Binary	0.80	0.87	0.89	0.77	0.83
Scheme 1 w/out OJ Ternary	0.58	0.79	0.69	0.81	0.66

Sens, Sensitivity; Spec, specificity; PPV, positive predictive value; NPV, negative predictive value; Acc, accuracy. For Scheme 0 and any ternary predictions, sensitivity, specificity, positive predictive value, and negative predictive value were averaged.

## Data Availability

The raw data supporting the conclusions of this article will be made available by the authors on request.

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
