# Peer review of "Artificial Intelligence for Predicting the Aesthetic Component of the Index of Orthodontic Treatment Need"

_bioengineering, 2024, doi:10.3390/bioengineering11090861_

Round 1
Reviewer 1 Report
Comments and Suggestions for Authors
The objective of this study is to use artificial intelligence (AI) to automate the AC assessment. 1009 pretreatment frontal intraoral photos with overjet values were collected. Each photo was graded by an experienced calibrated clinician. AI was trained using the intraoral images, overjet, and two different approaches.
what does this error speak about?-The discrepancy between the gold standard and the prediction 135 was back propagated to each layer of the network to update their parameters. Error! Ref-136 erence source not found. shows a schematic how the IOTN network was trained.
Also this one-To test the model, the AI was given 200 unique, new images with correspondin. C. Error! Reference source not found
Complete IOTN-AC distribution for 240 our sample can be found in Error! Reference source not found..
In the discussion chapter, there is only one bibliography. Please cite more recent published articles in this section.
- The manuscript is clear, relevant for the field and presented in a well-structured manner. The design is pertinent.
- The satistics of the study is well performed with relevant results.
- The manuscript is scientifically sound.
- The manuscript’s results are reproducible.
- Present the limitations of the present article.
- The conclusion is relevant.
- Also please perform a revion of english.
Comments on the Quality of English Language
Moderate
Author Response
Comment 1: “What does this error speak about?-The discrepancy between the gold standard and the prediction 135 was back propagated to each layer of the network to update their parameters. Error! Reference source not found. shows a schematic how the IOTN network was trained. Also, this one-To test the model, the AI was given 200 unique, new images with corresponding. C. Error! Reference source not found Complete IOTN-AC distribution for 240 our sample can be found in Error! Reference source not found.”
Response 1: We have fixed all the error messages related to the references and figures by deleting the erroneous text and highlighted the corrected areas where these errors previously occurred.
Comment 2: “In the discussion chapter, there is only one bibliography. Please cite more recent published articles in this section.”
Response 2: We have addressed this concern by citing five additional recent peer-reviewed articles in the Discussion section to support our findings and provide a broader context.

Reviewer 2 Report
Comments and Suggestions for Authors
Congratulations on your work. This could be a very useful tool. Its practical applications range from diagnostics and treatment needs to faster insurance verification, and overall quality control of services.
I have one concern that is not addressed in the text. You mentioned that the sample mirrored the US population according to race and that there could be some errors due to the manipulation of the pictures. Did you find any differences in errors between races? Given that the color of gingiva can vary, could this have any influence on the AI algorithm?
Additionally, could standardized photography contribute to a better algorithm by reducing the need for image tweaking?
This both could be added to limitations of the study.
Author Response
Comment: “I have one concern that is not addressed in the text. You mentioned that the sample mirrored the US population according to race and that there could be some errors due to the manipulation of the pictures. Did you find any differences in errors between races? Given that the color of gingiva can vary, could this have any influence on the AI algorithm? Additionally, could standardized photography contribute to a better algorithm by reducing the need for image tweaking? This both could be added to limitations of the study.”
Response:
Thank you for your insightful comments and questions regarding the potential influence of racial differences and standardized photography on our AI algorithm. Due to the limited time available for this revision, we were unable to conduct a thorough investigation into the relationship between errors and race. However, we acknowledge the importance of this consideration and intend to address it in our future research.
Our current study, with a sample size of approximately one thousand images, is relatively small by deep learning standards. We recognize that this limited dataset may affect the robustness of our algorithm. We agree that the variability in gingival color across different races could potentially influence the performance of our AI model. Standardized photography could indeed help mitigate some of these errors by reducing the need for image manipulation, thereby enhancing the algorithm's accuracy and reliability. We incorporated these points as limitations in our manuscript.

Reviewer 3 Report
Comments and Suggestions for Authors
This report shows a high potential of diagnostic tools based on AI. In my opinion, the study is properly performed and well written. I have a great respect for presenting true (even "nonperfect" or negative) results. Every part of the manuscript keeps a good scientific standard.
My only minor suggestion refers to scientific language:
instead of:
"From a patient or provider viewpoint, you do not want false negatives."
I would prefer a sentence similar to:
"From a patient or provider viewpoint, it is undesired to receive false negative scores."
Author Response
Comment: "From a patient or provider viewpoint, you do not want false negatives." I would prefer a sentence similar to: "From a patient or provider viewpoint, it is undesired to receive false negative scores."
Response: We have revised the sentence as per the suggestion: “From a patient or provider viewpoint, it is undesired to receive false negative scores.”

Reviewer 4 Report
Comments and Suggestions for Authors
Dear Authors,
The place in the article for cited references should be corrected because reference number 35 and 36 are in line line 216 and 32 - in line 295; 33 - 303, and 34 - 328.
Author Response
Comment: “The place in the article for cited references should be corrected because reference number 35 and 36 are in line 216 and 32 - in line 295; 33 - 303, and 34 - 328.”
Response: We have corrected the placement of the cited references accordingly.
